# Engagement with mHealth Alcohol Interventions: User Perspectives on an App or Chatbot-Delivered Program to Reduce Drinking

**DOI:** 10.3390/healthcare12010101

**Published:** 2024-01-02

**Authors:** Robyn N. M. Sedotto, Alexandra E. Edwards, Patrick L. Dulin, Diane K. King

**Affiliations:** 1Center for Behavioral Health Research and Services, University of Alaska Anchorage, Anchorage, AK 99508, USA; aeedwards@alaska.edu (A.E.E.); dkking@alaska.edu (D.K.K.); 2Department of Psychology, University of Alaska Anchorage, Anchorage, AK 99508, USA; pldulin@alaska.edu

**Keywords:** mHealth interventions, mHealth engagement, alcohol intervention, digital health intervention, chatbot

## Abstract

Research suggests participant engagement is a key mediator of mHealth alcohol interventions’ effectiveness in reducing alcohol consumption among users. Understanding the features that promote engagement is critical to maximizing the effectiveness of mHealth-delivered alcohol interventions. The purpose of this study was to identify facilitators and barriers to mHealth alcohol intervention utilization among hazardous-drinking participants who were randomized to use either an app (Step Away) or Artificial Intelligence (AI) chatbot-based intervention for reducing drinking (the Step Away chatbot). We conducted semi-structured interviews from December 2019 to January 2020 with 20 participants who used the app or chatbot for three months, identifying common facilitators and barriers to use. Participants of both interventions reported that tracking their drinking, receiving feedback about their drinking, feeling held accountable, notifications about high-risk drinking times, and reminders to track their drinking promoted continued engagement. Positivity, personalization, gaining insight into their drinking, and daily tips were stronger facilitator themes among bot users, indicating these may be strengths of the AI chatbot-based intervention when compared to a user-directed app. While tracking drinking was a theme among both groups, it was more salient among app users, potentially due to the option to quickly track drinks in the app that was not present with the conversational chatbot. Notification glitches, technology glitches, and difficulty with tracking drinking data were usage barriers for both groups. Lengthy setup processes were a stronger barrier for app users. Repetitiveness of the bot conversation, receipt of non-tailored daily tips, and inability to self-navigate to desired content were reported as barriers by bot users. To maximize engagement with AI interventions, future developers should include tracking to reinforce behavior change self-monitoring and be mindful of repetitive conversations, lengthy setup, and pathways that limit self-directed navigation.

## 1. Introduction

Given the negative health and social consequences of heavy drinking and the low reach of treatment, access to evidence-based treatment and support for excessive alcohol use is a public health priority [1,2]. Only around 8% of individuals with an alcohol use disorder receive any type of intervention [3]. Technology-based interventions have great potential to expand treatment access and have been shown to be acceptable among individuals who want to reduce their alcohol and or other substance use [4,5,6,7]. Interactive websites and mobile apps can be used to deliver direct treatment services [8], provide behavioral support for reducing alcohol use and prevent relapse [9], and have the potential to overcome personal and access-related barriers to in-person alcohol treatment. Common barriers include limited availability of services; cost and inadequate insurance; shame and stigma surrounding alcohol treatment; and challenges with leaving work, finding childcare, and obtaining transportation [10,11,12,13]. In addition to the potential to reduce barriers to treatment, technology-based interventions have also been shown to be effective at reducing alcohol consumption and reducing symptoms from other mental health concerns, such as anxiety and depression [14,15,16]. While these indications of effectiveness are encouraging, a challenge that is common to mHealth interventions relates to diminishing use over time, with numerous studies indicating low engagement may reduce effectiveness [17,18,19], as well as findings showing that high engagement is associated with increased efficacy [20].

With regard to mHealth apps, reported features found to enhance engagement include visually appealing and easily navigated content; reminders; ability to set goals, self-monitor, and receive feedback; reducing the number of actions users have to take when conducting the target behavior; providing tailored information to the user; rewards and incentives that encourage use; novelty, or regular content updates; and features that promote therapeutic alliance including encouraging acceptance, support, and relatability [21,22,23,24,25,26,27]. The therapeutic alliance between the participant and the intervention has also been shown to improve with the use of conversational agents that use artificial intelligence (AI) to guide and prompt participants and encourage general usage of apps, resulting in more engagement [28]. Engagement with technology-based interventions is measured through both behavior (frequency and amount of use) and experience (the user’s subjective experience, such as attention, interest, and affect) [24]. This paper reports on the subjective experience of engagement among users of the behavior change app Step Away, which was designed to reduce hazardous drinking.

### 1.1. Step Away: A Smartphone-Based App for Reducing Hazardous Drinking

Step Away is an app designed to provide a stand-alone, in-the-moment alcohol intervention that can be accessed anywhere the user has a data connection. It is oriented toward helping a user gain awareness of their alcohol consumption through daily assessment interviews and feedback on progress. Users interact with the app by setting goals, identifying their alcohol craving triggers, setting up a support system, and choosing non-drinking activities. Step Away and its veteran-focused version, Stand Down, have been shown to be efficacious in assisting individuals with reducing their problematic drinking [29,30,31]. A detailed description of Step Away functionality is available in a previous publication [29].

In prior studies, engagement with Step Away has been relatively high, with approximately 40% of participants continuing to use the app at 6 months; however, the extent to which participants engaged with all of the 8 modules was highly variable [29], thereby potentially diluting its effectiveness. To encourage full exposure to the intervention and promote participant engagement, we developed a chatbot version of Step Away. The chatbot incorporates modules from the app and delivers them in a conversational script. The bot and app both provide daily check-ins to track drinking behaviors and a menu of tools to support at the moment; however, a key feature that differentiates the bot from the app is the bot’s guidance to various modules of the intervention. The app relies on self-guidance from the user to engage with all available modules, while the chatbot provides a conversational interface to encourage the user to engage with all of the content while providing motivating reinforcement through the conversation. Research suggests that user guidance by the app leads to increased engagement [32], and the aim of the chatbot development was to increase user guidance through the newly developed chatbot. Figure 1 shows a screenshot of the dashboard of the Step Away app. Figure 2 shows a screenshot of a daily interview script with the chatbot.

### 1.2. Study Objective

The purpose of this paper is to report the results of qualitative interviews with study participants who were randomized to receive either the app or AI chatbot-delivered version of Step Away regarding their experience with the interventions and facilitators and barriers to their engagement. Our goals were to (1) obtain direct user feedback on facilitators and barriers to use; (2) understand which app and/or bot features motivated them to maintain use over time; (3) identify features that diminished their engagement; and (4) recommend ways to optimize mHealth engagement with AI and app features.

**Figure 2 healthcare-12-00101-f002:**
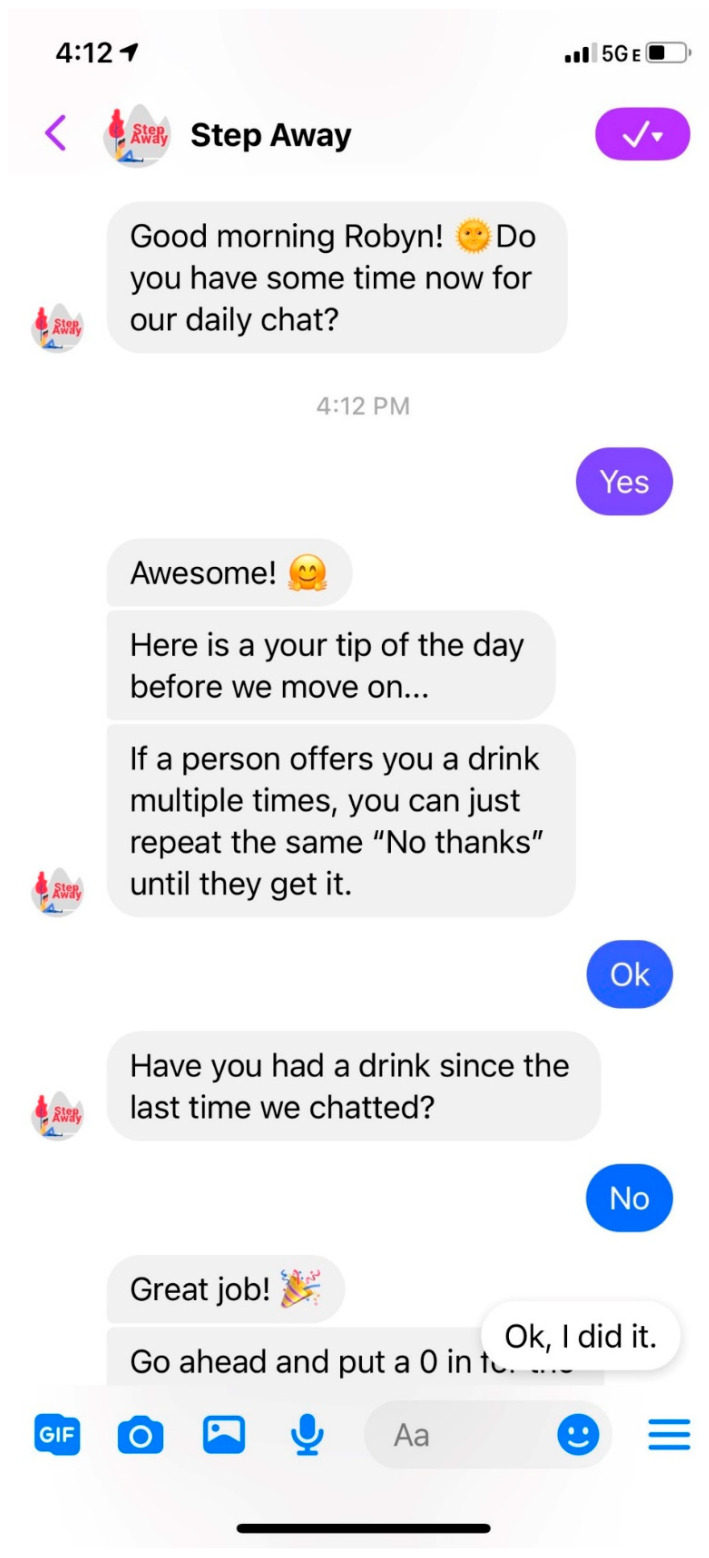
Sample conversation script with the chatbot.

## 2. Materials and Methods

### 2.1. Study Design

Details of the main Step Away study from which our participants were drawn are reported in a previous manuscript [20]. The previous manuscript reports on the efficacy and usage data from the study, while this current manuscript reports on qualitative data based on participant experiences with using either intervention. To summarize briefly, the main study participants were recruited via Facebook advertisements, were at least 18 years of age or older, were U.S. residents, were actively drinking, had English language proficiency, and had either an Android phone or an iPhone. Android is a trademark of Google LLC (Menlo Park, CA, USA), and iPhone is a trademark of Apple Inc. (Los Altos, CA, USA). Both products are registered in the U.S. and other countries and regions. Participants were also not currently enrolled in treatment or using another alcohol app. All participants reported drinking patterns that fell into the at-risk or high-risk levels, as determined by the USAUDIT, a 10-item screening measure to screen for hazardous drinking [33]. Participants were included in the study if their baseline USAUDIT scores fell between 8 and 24 for males and 7 and 24 for females or males over the age of 65. Participants with a possible AUD (i.e., USAUDIT scores of 25 or more) were not eligible to participate and were provided treatment locator resources.

In the main study, participants (*N* = 191) were enrolled and randomly assigned to either (1) utilizing the Step Away app (for iPhone and Android smartphones), (2) utilizing the Step Away chatbot (provided on Facebook Messenger), or (3) an assessment only control group. Participants were assessed at baseline and at a 12-week follow-up on their alcohol consumption and related behaviors. After the follow-up assessment was completed, the control condition was offered to utilize the intervention. At follow-up, 150 participants were retained for analysis. Within the main study, we found that at 12 weeks, participants in both intervention groups reduced their drinking substantially, with large effect sizes noted. Participants in the waitlist control condition also reduced drinking substantially over the course of the study; results regarding the effectiveness of the app and chatbot relative to the control condition were thus inconclusive [20]. Secondary results indicated that engagement with either intervention was significantly associated with a reduction in various drinking measures. Duration of use, or the length of time the participant engaged with either intervention, was associated with a significantly greater increase in the percentage of days abstinent at follow-up and a significantly greater reduction in drinks per drinking day. We collected qualitative data from participants who used the interventions to identify features of the app or chatbot that enhanced usability and encouraged or diminished their continued engagement.

### 2.2. Data Collection

Twenty semi-structured telephone interviews were conducted with participants. These 20 participants were recruited from the 105 study participants who utilized an intervention, either the bot or the app, and completed follow-up assessments. Participants were purposefully selected to maximize variation in usage of the app or the chatbot. To ensure all interview participants had downloaded and tried out the app or bot, the minimum threshold of app or bot use to be selected for an interview was set at 10 sessions. We recruited 10 participants from both intervention groups, consisting of 5 participants who utilized the intervention the most and 5 who utilized the intervention the least, totaling 20. We anticipated this would be an adequate sample to maximize the diversity of perspectives and still reach saturation on themes.

The 55 app users were organized in order from least to highest app usage. A total of 20 app users who utilized the app the least and 20 who utilized it the most were marked. From the 20 ‘high users’, 5 were randomly selected. From the 20 ‘low users’, 5 were randomly selected. The same process was repeated with the 45 bot users, randomly selecting 5 ‘high users’ and 5 ‘low users’. These participants were contacted via email and telephone to schedule their interview time. Three participants who were originally selected did not answer the three contact attempts for interviewing and were replaced with three randomly selected participants from their respective intervention and usage groups.

Interview questions were developed from the results of the follow-up survey open-ended responses. There were 11 core interview questions that were asked of both app and chatbot users, plus an additional chatbot user question about their experience with and impressions of the chatbot’s conversational quality. These questions were reviewed and pilot-tested for comprehensibility. Interviews took place in December 2020 and early January 2021. The average interview was approximately 20 min. Participants were compensated with a USD 25 e-gift card for completing their interview.

### 2.3. Analysis

Thematic analysis was used to analyze data from the 20 telephone interviews. Interviews were audio-recorded and transcribed using an automated transcription service. Participants’ names were replaced by a researcher-selected identification code; no identifiable information was audio-recorded. Transcripts were then reviewed by the research team for accuracy. A comprehensive set of a priori codes was identified and auto-coded using NVivo version 12. These a priori codes were developed based on specific questions asked during the interview (e.g., overall experience, preferred recommendations given by the app/bot, features that increased engagement, features that decreased engagement). Two members of the research team first reviewed the transcripts independently. The secondary analysis method of content analysis was used to identify new themes that emerged from the data other than the a priori themes. The two research team members reviewed transcripts in sets of 5 and met to discuss themes that emerged after each set. When discussing, the team members clarified code definitions and continued this process until all 20 interviews were reviewed and all themes that emerged were discussed. The two team members constructed a codebook, which was reviewed with the rest of the team, and constructed maps to better understand how the themes related to one another and which themes mapped more strongly to the bot user experience versus the app user experience. After the codebook was completed, the research team members reviewed the interviews one more time to ensure that no new codes emerged and that all text had been coded appropriately. Table 1 shows the telephone interview questions.

### 2.4. Ethics Approval

The study was reviewed and approved by the University of Alaska Anchorage Institutional Review Board 1521800-2 on 20 December 2019.

## 3. Results

### 3.1. Sample Characteristics

Table 2 shows participant characteristics by demographic. Of the 20 interviewed participants, the mean age was 43.45 (SD = 13.74), and ages ranged from 28 to 68. Slightly over half of the sample identified as male. Over half of the sample (75%) had a bachelor’s degree or higher.

### 3.2. Themes

Major themes that emerged are organized by features that promoted app or bot engagement, features that diminished app or bot engagement, and participant recommendations for enhancing the app or bot. Table 2 shows interview participant demographics. Themes are listed in Table 3 and grouped as being more salient for app or bot users or themes that emerged equally in both groups.

#### 3.2.1. Features That Promoted Engagement in Both Groups

Features of both the bot and the app that were identified as promoting continued engagement were tracking, increased insight, notifications, daily recommendations, feedback, accountability, personalization, and positivity. Themes are listed in order of frequency.

##### Tracking

The most commonly mentioned feature that promoted and sustained engagement for both groups of users was the tracking feature. Respondents stated that the tracking feature allowed them to obtain feedback, accountability, and work toward their personal goals. When asked what feature was the most helpful, one participant stated, “One hundred percent the tracking was the most helpful, reminding me to track, having me think about it…helped me be more consistent in not drinking or drinking less.”

##### Insight into Patterns

Users of both the bot and the app reported that the interventions helped them become more mindful of their drinking habits and gain insight or awareness into their drinking. A bot user stated, “*I hadn’t realized that I was doing things just out of habit…it made me take a look of when and why I do what I do instead of just kind of being on autopilot.*”

##### Notifications

Participants indicated that receiving daily notifications about high-risk drinking times and reminders to track their drinking encouraged them to maintain their engagement with the intervention over time. An app participant mentioned how this was helpful: “*I think one of the most helpful things that I found was…the daily reminders of, ‘Hey, a high-risk time is approaching consider doing an activity you enjoy’.*”

##### Daily Tips and Recommendations

Participants referenced the daily recommendations of strategies and tips to reduce drinking as helpful. A bot participant mentioned using these tips: “*I really liked the…daily tip, how every time when I would open it, it would come up with that and I would usually try to take those to heart and use those tips.*” An app participant stated the helpfulness of these reminders as well: “*having water in there, you know, drinking something before you go out to a party— there were some tips in there that popped up that were good.*” This theme emerged for both participants but was stated more frequently by bot users, potentially due to the bot-directed conversations about daily tips.

##### Feedback

Another engagement-promoting feature identified by app and bot participants was the feedback provided by the app or bot to the user about their drinking. This feedback was in the form of graphic displays that showed the user how much they were drinking and progress toward meeting their goals. One app participant stated that this graph raised their awareness: “*I think for me, the biggest benefit is just being able to track in real time, how many drinks I’ve had and go back and look and see—have it graphed out…weekly, daily, so I can look and see.*” While feedback emerged as important for both app and bot participants, the description of the graph allowing users to view their own progress emerged more strongly for app participants.

##### Accountability

Participants in both the app and bot groups stated that the interventions provided a sense of accountability for them in their goals to reduce drinking. One bot participant stated that this encouraged them to continue working toward their goal: “*it was a nice habit or what felt like an accountability friend to keep me in control of my drinking.*”

##### Personalization

Participants mentioned that personalization features were helpful, such as being able to set a personalized goal. A bot participant mentioned the ability to personalize drinking goals: “*I’m really grateful that it’s something that’s being created because I think it has a lot of potential to help a lot of people who don’t necessarily fall into the ‘I need to go to AA category, but I do need to reduce my intake for my health’.*” An app participant also mentioned the personalization of moderating their drinking: “*I think an app is the wave of the future for something like this. I mean the other tools out there for people, like AA, or for somebody like me that just wanted to cut down. I didn’t want to quit. There’s not a lot of resources out there for somebody who only wants to drink four beers at night instead of six beers a day.*” Another bot participant stated the bot became more personalized with more use “*as I went on with the program longer and started spending more time with it and entering more information, I could definitely tell that it was doing a better job at personalizing things for me.*”

##### Positivity

Both app and bot users reported they felt the interactions were positive and uplifting. Bot users discussed this theme in the context of the conversation and how the bot was positive and encouraging in its responses. App users mentioned the positive reinforcement they received from meeting their goals. One app user stated they felt this positive reinforcement when tracking their drinks: “*every time you answer and it’s below, it’s like, ‘Hey, hurray, you’re one more day closer to your 30-day goal’.*” The theme of positive conversation emerged strongly for bot users. A bot participant stated, “*I like the positivity of it. It was always a positive response.*” Some bot participants also indicated they felt a connection with the chatbot. A bot user said, “*it was reminding me daily, you know, I was like, ‘Hey, you know, chatbot’s going to check in with me’.*”

#### 3.2.2. Features That Diminished Engagement

##### Technology Glitches

Technology glitches were frequently mentioned by all users as a barrier to use. Such glitches included problems with the setup process, notifications, and ease of navigation to various features. One participant described a point in which their data seemingly disappeared: “*It just threw away all of my data and I’d had it for a month. I was really into it…and then when it said, ‘Oh, I’m sorry, let me go ahead and reregister you’, all my data was gone.*” A user also indicated there were technology glitches with the app when tracking drinking: “*on my weekly summaries, it would indicate that I reached my goal, and I hadn’t.*”

##### Notifications

Both app and bot users described difficulties with the notifications of both programs; however, this was a more central theme for bot participants. When discussed by app participants, the issues with notifications mainly centered around the prompts not arriving at the time the user had requested to receive them or that prompts were too close together in time. An app user stated, “*I still get double reminders…I get two reminders at 9:00 P.M. for a daily interview and I get two reminders at 10:00 P.M. for a daily interview, even if I’ve already done it.*”

For bot users, issues with notifications were more related to not receiving any. The bot was hosted through Facebook Messenger, which had unique limitations regarding sending notifications. If a user did not respond to a prompt to engage in an interview, the messenger bot would not provide notifications on subsequent days. Bot participants expressed this: “*…and then the second thing was that if you didn’t respond to one of its prompts, it would, it would just sit there.*” Another bot participant mentioned how this prevented their use of the bot: “*with a Facebook message…if I’m not able to respond in the moment, I will forget about it. And then once I don’t respond for long enough, then the chatbot kind of goes dormant. So, it was just hard for me to kind of keep the engagement active.*” The lack of notifications for bot users prevented continued use of the intervention.

##### Feature Usability

Users mentioned features of the app or bot that interfered with usability, such as not understanding how to change their goal or selected strategy. An app user mentioned that entering drinking data required scrolling to select answers that could have been more easily filled in: “*It was either the time or the dates and you had to use a wheel…It wouldn’t just let you input the numbers. So, it made it very cumbersome, especially in the start where you had to put in for 30 days…how many drinks you had or something like that.*” Another participant mentioned difficulty entering missed data in the follow-up survey: “*if I forgot to put in data for about a week it wouldn’t give me the dates…it was asking me to input number of drinks but said something like* “*last Tuesday*” *and* “*Tuesday*” *and I wasn’t sure which Tuesday it was referring to.*”

##### Repetition

App and bot users discussed repetition in the prompts or strategies suggested. Bot users, in particular, indicated that responses were repetitive and overly scripted. One bot user stated, “*Depending on what it asked, and [what] you said, it was the same verbiage every time.*” App users mentioned that the tips or modules were repetitive. An app user stated, “*…yes, I did use several of them at the beginning, but…towards the end I kind of just was like, ‘I get it’.*” Another app user discussed repetition with the recommended strategies: “*when I had a chance to pick the strategies and every day it was like drink water between beverages. So, if you’re only having two, it’s one glass of water, like it just, it was no longer an effective tool.*”

Users also discussed how the recommendations and tips by the app or bot were insufficiently tailored or relevant to their situation. A participant using the bot described how the recommendations were not helpful to their situation during the pandemic: “*You know, the, the stress of COVID was not going to be alleviated by the things that were in that chatbot. Like go get a drink of water, that’s not going to solve my stress problem.*”

##### Setup Process

Some participants mentioned that the setup process for both the app and the bot was cumbersome or lengthy. An app participant stated they felt they were answering many questions during setup: “*Every button has five more buttons underneath it…and so, it just seemed like I was just taking survey after survey…at least twice a week.*”

#### 3.2.3. Features That Diminished Engagement Unique to Bot Users

No themes for barriers unique to the app emerged, but a few emerged that were specific to the chatbot. The themes of chatbot-specific barriers were the Facebook platform, conversation quality, the lack of self-direction, and app preference.

##### Conversation Quality

Bot users discussed how the conversation with the bot prevented their use of it. One participant stated, “*The script was pretty…clunky. It wasn’t adaptive.*” Bot users discussed how they primarily wanted to track their drinks and not hold a conversation with the bot for each interaction with it.

##### Lacking Self-Direction

To utilize the bot, the user had to engage in a conversation. The bot would follow a script and guide users to track their daily drinks or complete a module for the day to learn about strategies to reduce their drinking. Use of the bot was thus guided by this conversation, whereas use of the app was self-directed by the user. Participants of the bot group indicated that this lack of self-direction acted as a barrier to their use of the bot. One bot participant described how he wished the drink tracking process could be more streamlined: “*I wish you could just get straight to the calendar without answering*.” Another bot participant mentioned that the bot would recommend modules that they were not ready to complete: “*I like to go through things like pretty methodically, so it kind of recommended something when I wasn’t necessarily like ready for it or in a place to do something with it at that moment*.” Another bot participant mentioned that they were more accustomed to the self-direction an app provides: “*I’m used to using an app so I can login whenever I want and I record things whenever I want but having a chatbot kind of bother me on a regular basis was, it was strange*.”

##### Facebook Messenger Platform

The chatbot was hosted through Facebook Messenger, which some participants reported as a barrier to their use due to low engagement with Facebook. As mentioned in the notification theme, Facebook experienced technology glitches in sending out daily notifications. These glitches were more frequently reported by the group utilizing the bot. Table 4 shows descriptions of each of the features that dimished engagement. 

#### 3.2.4. Recommendations for Optimizing Utility and Engagement

The main themes suggested by participants of changes to the app or bot coincided with the barriers experienced by both.

##### Personalization

Participants suggested ways to increase the personalization of the app or bot. An app user stated they would appreciate being able to interact with the system more about their specific goals: “*if it was really sophisticated, I could have put in goals and, and it would have…done more to kind of talk to me about the goals I was trying to aim for.*”

##### Fix Technology Glitches

Participants mentioned that the app or bot would be greatly improved by fixing technology bugs and notification issues that were expressed as barriers to use. An app participant referenced their difficulty with losing their data: “*if it could store data somewhere so that you didn’t have to re-enter all that stuff or if somehow that didn’t ever happen again.*”

##### Enhance Usability

Usability changes were suggested by both the app and the bot participants. Some participants suggested the ability to make corrections to data they had entered. An app user suggested another usability change to the way drinks were entered: “*when it asked you for the time of day that you started and stopped…they should just have like 15-min increments, not one-minute increments. So I, you know, nobody’s going to put it in there, ‘I started drinking at 5:43 P.M.’, you know. It’s either 5:15, 5:30, 5:45 around the hour.*”

##### Connect to Resources

Participants suggested the ability for the app or bot to connect to local resources or support. An app participant suggested phone calls: “*having a weekly phone call or something, whether that’s a phone call from a human or just like a robo advisor.*” A bot user suggested that connecting to local counseling resources would also be helpful: “*If it comes to a question that it cannot answer, you can say, ‘Hey, I don’t know if I can quite answer this, let me give you a list of providers in your area’…or give you a connection either to…a licensed AODA counselor who you can chat remotely with.*”

##### Increase Accountability

Participants suggested improvements by increasing the accountability features. A bot user suggested doing so by increasing the reminders if the user has not interacted recently: “*There needs to be a greater accountability piece on part of it. I also use an app called MyFitnessPal…and it will give me a reminder it’s like, ‘Hey, you haven’t checked in with me, in three days, what’s up with that?’…that’s one thing that I think would be more helpful.*”

##### Feedback Changes

Both bot and app participants suggested improvements to the feedback provided by the bot or app. Some participants felt more visuals would be helpful. One participant indicated that they would like to have the ability to see more of their progress over time through a larger graph: “*It would be good to have a way to select the kind of timescale graph. Like, you know, you want to look at the stock market, just an overall pattern…they would take it over months.*” Another participant suggested a zoomed-out calendar: “*I was hoping at the end of it, um, that they would send a thing for like the three months that you did it.*”

##### Conversation Quality (Bot Specific)

Bot users mentioned the barrier of conversation quality with the messenger bot. They suggested increasing the conversation quality by including variety in the responses. A bot user suggested “*varying the responses and make it feel not as much of a program.*” Other users requested fewer affirmations: “*I think some of the affirmations, they got a little bit too chatty…it was a little too affirming for me.*”

##### Platform Change (Bot Specific)

Bot users suggested a different platform that is not Facebook Messenger. One user said, “*I think I would have preferred to see it on a different platform because I don’t use…Facebook messenger as often as I used to and sometimes I struggle with getting the notifications from Facebook.*” Another suggested, “*It could like be cross plat-, you know, like work along with texts or other types of notifications beyond just Facebook messages. That, for me, was the challenge just because of the Facebook limitations.*”

##### Combine the Features of the App and Bot

A theme for suggested changes that emerged was combining the app and bot in the future to have one intervention that holds the positive aspects of both. A bot participant mentioned that the conversational aspect of the bot would be combined with the self-direction of the app:

“*I think having the ability to have to be signed up in chatbot was really nice, I like that daily check-in, and if there was like a thing that connected you, like when you clicked on a link for it…it took you to the app, so you can kind of do either style. You can do the daily check-in, but if you’re a person who likes to go through things a little more, like by the book in a specific order, you can, you could still access that. So I think like an interface where you have the bot and the app connected to each other, so that you can kind of go between the two based on whether you need to be prompted or whether you’re saying…I’m going to make a choice to go work through some of these things a little more in depth.*”

Table 5 shows themes of recommendations. 

## 4. Discussion

### 4.1. Lessons Learned

This study focused on determining barriers and facilitators of engagement with an app or chatbot-based intervention for alcohol use reduction. Results support the previous literature that providing feedback, notifications, and personalization features are motivators for engaging in behavior change apps, while technical difficulties create barriers to engagement [26]. Participants in both intervention groups reported that the features most important to their continued engagement were those that supported their ability to track their drinking and monitor their progress, prompts and reinforcing messages from the app or bot, and self-assessments and feedback that increased their awareness of their drinking patterns and helped them to be more mindful about their drinking. Users reported that the notifications to check in with the app or bot encouraged their engagement, and when there were technological errors with these notifications, they engaged less. Overall, participants reported appreciating the ability to moderate their drinking instead of abstaining and to work on reducing their alcohol consumption without going to in-person, traditional forms of treatment.

This study adds new information related to engagement with a chatbot version of the app. The Step Away chatbot was created to facilitate engagement with the intervention, and there were lessons learned from this pilot study on how the chatbot provided its own unique facilitators and barriers to utilization. A barrier that was unique to bot users was the lack of self-direction the bot script provided the opportunity for. Participants indicated a desire to utilize the parts of the bot they wanted to use, such as tracking their drinks, without responding to the full script first. This supports previous research suggesting that continued guidance after setup with the app may be a nuisance to users [32] and that reducing the actions users need to take to implement the target behavior is a facilitator of engagement [27]. Additionally, the repeated bot conversation during the daily interview feature may have hindered utilization: many bot participants indicated that the conversation script felt rote or repetitive. Bot participants were divided on whether they appreciated the affirmations given by the bot, with some stating these affirmations were motivating and others reporting they felt the conversation was too affirmative. As tracking drinks and receiving visual feedback emerged as such a strong facilitator, it is recommended to other developers that, if utilizing a chatbot script, allow users to access these primary features without having to engage with the bot.

Consistency in providing outreach and prompting users, a hypothesized advantage of the chatbot, fell short due to platform constraints. Numerous participants mentioned using the bot less because they were no longer being prompted daily, while the app users were prompted regularly regardless of their response to the notifications.

One user suggestion was an intervention that combines the features of the app and chatbot into one. Combining preferred features of the two interventions would allow for more self-direction from the user to engage in tracking, monitor their own progress, receive feedback, and engage in positive dialogue. Combining the app and bot would also remove some barriers, such as the Facebook platform, repetitive conversation, and ill-timed prompts. Participants could have the option of speaking to the bot but would not be required to converse with it to access various features of the intervention. Having the chatbot included could also provide some conversational assistance with setup and introduction to the key features, as well as built-in opportunities for positive conversation, reinforcement of progress, and connection. It is possible that a combined app and bot intervention would lead to fewer technological barriers and more engagement with preferred features that are sustained over time.

Finally, the COVID-19 pandemic had varying effects on participant utilization of either intervention and varying effects on participant alcohol consumption. Some reported they used the app or bot less because they were drinking less: these participants referenced social drinking as why they were drinking less, stating they had fewer opportunities for social drinking during stay-at-home orders or establishments closing. Another reason participants stated for using the intervention less during the pandemic was the stress the pandemic caused, making reducing their drinking less of a priority. Others reported using the intervention more, referencing increased drinking or increased free time as their reasons.

### 4.2. Recommendations for mHealth Apps

The feedback from users of the bot and app highlights essential recommendations for designers and researchers in the field of mHealth behavior change interventions. Users consistently emphasized the importance of three key aspects: (1) notifications to encourage continued use; (2) ability to easily track the target behavior; and (3) feedback with progress reporting on the tracked behaviors. Incorporating visual aids such as graphs that display progress toward goals over time is strongly recommended. Additional feedback mechanisms that enhance user insights may include weekly reports on the target behavior and feedback on behavior patterns that illustrate progress over several months.

Notifications also emerged as a critical factor driving user engagement. It is advisable to develop personalized notifications that align with the user’s preferred notification schedule. As users in this study appreciated the ability to self-direct their usage of the app or bot, an easily accessible menu and settings feature is encouraged.

Another noteworthy recommendation is the regular rotation and updating of content to prevent it from becoming repetitive. Incorporating large language models (LLMs) like GPT-4 for more sophisticated and context-aware interactions could be considered, providing a more dynamic and responsive experience to users. Additionally, allowing the intervention to connect users with online educational resources or local support networks, such as group meetings or counselors, can enhance the overall user experience and effectiveness of the intervention.

While this study focused on the development of an alcohol intervention app, the insights into facilitators and barriers to use are likely applicable to other mHealth apps aiming to foster behavior change. User engagement remains a critical factor for the efficacy of mHealth apps, and these findings can inform the design and promotion of engagement in various mHealth interventions.

## 5. Limitations

The study sample was recruited through Facebook advertising, which may have limited the sample to participants to demographic groups who use Facebook over other social media platforms. Potential participants may also be more technology savvy than the general population of at-risk drinkers. The participants who were interviewed were highly educated, with the majority having a bachelor’s degree or higher. This may have influenced their experience with either technology. Additionally, those who participated in the interviews were engaged enough to respond to the email or phone call, recruiting them for the follow-up interview. There were a few participants who were randomly selected that did not respond to the interview recruitment. While we attempted to hear from differing perspectives by recruiting both high and low utilizers, the participants who completed the interviews may have had differing experiences than those who did not respond to the interview recruitment emails and calls. Due to time and resource constraints, the sample size was limited to 20 interviews; however, by purposeful selection based on utilization, we were able to obtain diverse perspectives.

The chatbot was also in an early stage of development and had a few unfortunate glitches that users reported to be barriers to continued engagement. Other more refined chatbots may elicit different feedback from users than what is described in this study. This feedback reported from users is not necessarily representative of chatbots in general.

## 6. Future Research

There were features specific to both the bot and the app that facilitated use and features from both that hindered use. Participants endorsed an interest in a combined version of the chatbot and app that balanced proactive reminders and encouragement with the ability to self-navigate and track progress against goals. Future research may wish to combine these interventions into one intervention and test if this combination results in increased utilization. In the development of other digital behavior change interventions, researchers may wish to research the combination of a bot or messenger that is available to guide users through an app’s features. Future qualitative research should also work to recruit a larger sample size to garner more diverse perspectives. This chatbot was also only accessible on Facebook Messenger, which some participants reported to be a barrier. Future development may circumvent this barrier through the use of another bot platform. Future development may also wish to include a social support component to connect with other users.

Engagement is a growing concept in research, given its association with intervention outcomes; however, it can be measured in various ways, and researchers have encouraged the priority of improving engagement measures [34]. While our study explored facilitators and barriers to utilization to explain differences in utilization of Step Away [35], focusing on utilization alone is likely insufficient to determine whether the skills learned through the intervention were incorporated into the user’s everyday “offline” life [34]. This may be measured by qualitative data regarding the user’s application of the skills learned and how they used advice from the app, as well as data collected within the app on how users were able to apply these skills. A deeper understanding of the minimum usage associated with significant behavior change may also guide future studies with alcohol intervention apps.

## 7. Conclusions

This study compared two mHealth interventions aimed at alcohol reduction: a smartphone app, Step Away, and its newly developed chatbot version. Participants were interviewed on their experiences with the app or the bot. Many felt that the ability to track their drinking, obtain feedback, and receive notifications to engage with the app or bot encouraged their continued usage of either intervention. There were barriers unique to the bot and facilitators unique to the app that most likely led to more sustained utilization of the app. Participants using the app reported appreciating the ability to self-direct their use of the intervention, while participants of the bot reported they wished they had more of an ability to self-direct their usage. Participants indicated they would be interested in a combined version of the app and bot, which may increase facilitators’ engagement and reduce barriers.

## Figures and Tables

**Figure 1 healthcare-12-00101-f001:**
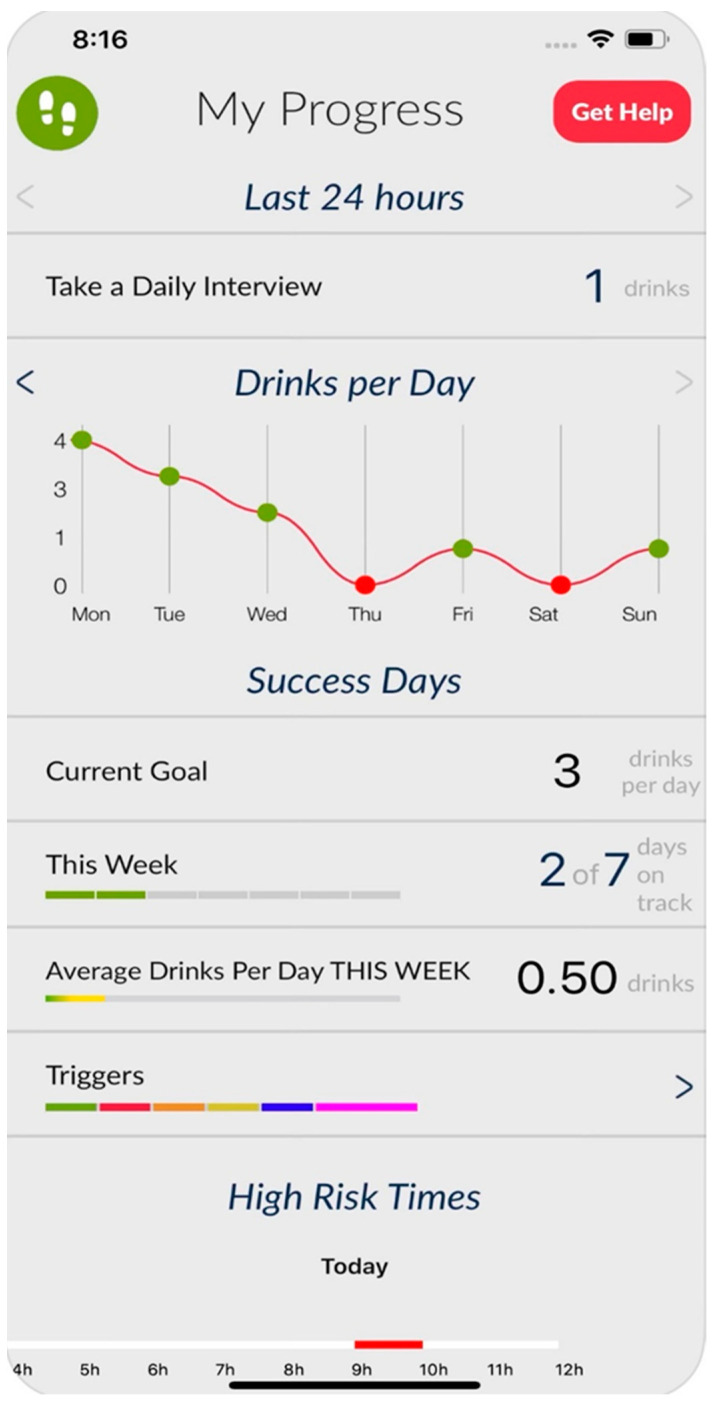
Main screen of the Step Away app.

**Table 1 healthcare-12-00101-t001:** Interview questions.

1	How would you describe your overall experience with the Step Away app/bot?
2	What did you learn about drinking or changing drinking from the Step Away app/bot?
3	The app has steps or mini-modules that provide information and strategies related to changing drinking. Did you have the opportunity to use them? *Probes*: What were your impressions of them? Which specific feature did you find was the most helpful or supportive of your goals? Which specific feature did you find was the least helpful or supportive?
4	For some things, like the daily interview, the Step Away app/bot reaches out to you, and for others, like the activities step, the Step Away app/bot needs you to open it. Would you have preferred more prompting or reminders to help connect you to activities or other features?
5	Was the app/bot able to anticipate your drinking triggers? *Probes*: What sorts of recommendations did it make to help you manage them? Which recommendations did you prefer? Did you use any of these recommendations?
6	To what extent did you feel the Step Away app/bot provided you with personalized support and suggestions to help you achieve your specific goals? *Probe*: Do you have an example?
7	*For the regular users*: What is it about the Step Away app/bot that makes you want to keep using it? *For the infrequent users:* What is it about the Step Away app/bot that kept you from using it more often?
8	To what extent did having human contact, for example, receiving a phone call or email from researchers in the study, encourage you to keep using the app/bot?
9	To what extent did the pandemic factor into your usage of the app/bot? *Probe*: Do you think you would have been likely to use it more/less/not at all?
10	If you could make changes to the app/bot what would they be? *Probe*: Why?
11	Do you have any other thoughts about the Step Away app/bot that you would like to share?
12	*This question was asked only to chatbot users:* One of the benefits of a chatbot is its ability to have a conversation with you. What are your thoughts about the conversation quality of the chatbot? *Probes*: For example, did the conversation feel natural or like you were talking to a real person? How do you think the conversation quality could be improved? What about the chatbot’s conversation did you enjoy? Did you feel connected or a relationship developing?

**Table 2 healthcare-12-00101-t002:** Participant demographics.

Measure	Mean (SD)
Age, mean (SD)	43.45 (13.74)
21–29	2 (10%)
30–39	8 (20%)
40–49	3 (15%)
50–59	3 (15%)
60–69	3 (15%)
Gender, n (%)	
Male	11 (55%)
Female	9 (45%)
Race, n (%)	
Black or African American	1 (5%)
Caucasian	17 (85%)
Asian American	1 (5%)
Hispanic or Latinx	1 (5%)
Education, n (%)	
Some college	3 (15%)
Associate’s degree	2 (10%)
Bachelor’s degree	8 (40%)
Master’s degree	5 (25%)
Doctorate	2 (10%)

**Table 3 healthcare-12-00101-t003:** Features that promoted engagement.

Feature	Description	# of References (App, Bot)
Tracking	Ability to record number of drinks per day	(32, 22)
Insight	Feedback from the app or bot and tracking drinking helped increase awareness about participant drinking behavior	(8, 21)
Notifications	Reminders from the app or bot to track drinks or engage with the intervention	(15, 13)
Daily tips	Daily suggestions of coping skills or strategies to reduce drinking	(5, 17)
Feedback	Graphs of drinks per day and ability to see goals met over time	(8, 4)
Accountability	Having the app or bot as a reminder of one’s goals	(5, 5)
Personalization	Being able to customize features of the app or bot to one’s goals or preferences	(2, 6)
Positivity	Encouraging conversational tone or affirmations	(1, 4)

**Table 4 healthcare-12-00101-t004:** Features that diminished engagement.

**Both Bot and App Participants**		
Feature	Description	References (App, Bot)
Technology glitches	Reported glitches with either intervention included notifications not working properly, deleting data or progress, or displaying incorrect data	(11, 11)
Notifications	Notifications not working in frequency or at time of day that was requested or expected	(7, 10)
Usability	Any mention of usability features needing improvement or hindering use	(5, 9)
Repetition	Content or responses being too repetitive	(4, 7)
Setup	Reported the setup process was lengthy or glitched	(6, 1)
**Bot-Only Participants**		
Feature	Description	References (App, Bot)
Conversation quality	Conversation scripts given from the bot messenger discussed as repetitive or rote	(0, 11)
Lacking self-direction	The bot script guided participants through the drink tracking and steps, removing user ability to direct their experience	(0, 7)
Facebook platform	Facebook Messenger was the platform used for the bot. Some participants referenced preference to not use Facebook or glitches specific to Facebook Messenger.	(0, 3)

**Table 5 healthcare-12-00101-t005:** Recommendations.

**Both Bot and App Participants**		
Recommendation	Description	References (App, Bot)
Personalization	Suggestions for how to customize the bot/app more included adapting the conversation script, more options for notifications, and more educational resources, including health-related resources	(8, 12)
Fix glitches	Fix referenced technology glitches (e.g., data storage, notification issues)	(6, 8)
Enhance usability	Suggestions to change to how drinking data are entered, the layout of the app/bot, and changing previously selected personalization features	(4, 4)
Connect to resources	Suggestions to provide more connections to local providers in the community, discussion with a live person, or additional educational resources	(2, 5)
Feedback changes	Suggestions to improve the visual feedback by including graphs with a longer timeline	(1, 2)
Increase accountability	Increase accountability by sending unique notifications to users who have not been engaging with the intervention	(0, 2)
**Bot-Only Participants**		
Recommendation	Description	References (App, Bot)
Conversation quality	Bot participants provided suggestions of how to increase conversation script quality (e.g., less repetition, more personalization, fewer affirmations)	(0, 10)
Platform change	Suggestion of providing the bot through another platform than Facebook Messenger	(0, 3)
Combine app and bot	Adding the conversational bot to the app was suggested	(0, 1)

## Data Availability

Data are contained within the article.

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
