# Peer review of "Engagement with mHealth Alcohol Interventions: User Perspectives on an App or Chatbot-Delivered Program to Reduce Drinking"

_healthcare, 2024, doi:10.3390/healthcare12010101_

Round 1

Reviewer 1 Report

Comments and Suggestions for Authors

Engagement with mHealth alcohol interventions: User perspectives on an app or chatbot-delivered program to reduce drinking

This is useful up-date of the previous up that provide better engagement and accompanying the patients (or users) in the process of dropping (reducing) alcohol consumption.

Some notes to be considered for improving the manuscript:

Abstract // Methods

1.       You mentioned only N for the interview; however, for other analysis types you have N =191. It is good to reflect the total sample and methods description.

2.       How these 20 were selected and assigned to the subgroups? (this is more suitable to describe in the Methods part)

3.       Line 182: please, provide an average interview duration time).

4.       Line 204: For Ethics board approval, it is good to put the date also and N if exist.

Results:

5.       Table 2 (demographics): It is good also provide the distributions for each subgroup

The last observation concerns not to this study, but maybe “Future development directions”. Since users registers can be also anonymous (I guess), participating in the “Groups results comparison” or including other users groups chats could be also a good perspective of encouraging and support.

Reviewer 2 Report

Comments and Suggestions for Authors

Thank you for the opportunity to review this manuscript on digital solution as an intervention to reduce alcohol consumption.

Please find my suggestions and questions.

MANUSCRIPT

-The objectives must be reviewed:

·       Abstract

Lines 12-15: “The purpose of this study was to identify FACTORS related to mHealth alcohol intervention utilization amongst hazardous-drinking participants who were randomized to use either an app (Step Away) or Artificial Intelligence (AI) chatbot-based intervention for reducing drinking (the  Step Away chatbot).”

·       Introduction (The objectives were described twice in this section)

Lines 65-66: “This paper reports on the subjective experience of engagement among users of the behavior change app, Step Away, which was designed to reduce hazardous drinking.”

Lines 126-131: “The purpose of this paper is to report the results of qualitative interviews with study  participants who were randomized to receive either the app or AI chatbot-delivered version of Step Away. Our goals were to: (1) obtain direct user feedback on facilitators and barriers to use; (2) understand which app and/or bot features motivated them to maintain use over time; (3) identify features that diminished their engagement; (4) recommend  ways to optimize mHealth engagement with AI and app features.”

·       Material and Methods  (The objectives were again described in this section)

Lines: 158-162: Our objective for collecting qualitative data from participants who used the interventions was to identify features of the app or chatbot that enhanced usability and encouraged or diminished their continued engagement, to improve effectiveness of mHealth delivered alcohol interventions.

ABSTRACT

-Please, describe the date of the study.

INTRODUCTION

-Starting from line 65, the text is confusing. The authors wrote about the objectives of the manuscript. Next, the authors cited a study on Step Away usage. Subsequently, the authors described the Step Away application.

-Please, the title of the figure must be described below it.

MATERIAL  AND  METHODS

The description in the "Material and Methods" section is confusing. The authors carried out a previous randomized study, using the Step away app and the Step away chatbot as interventions. The quantitative results have already been published. This has to be summarized and understandable. At this point, the authors intend to present the perspectives (qualitative results) of some participants from these intervention groups, and this part must be well described.

-Lines 134-135: “Details of the main Step Away study from which our participants were drawn are reported ELSEWHERE.” Suggestion: Instead of the authors writing "elsewhere", they could write “in a previous study”, for example.

-Line 142: Please, describe USAUDIT.

-Lines 146-154: “Participants (N=191) were enrolled and randomly assigned to one of three groups: Step Away app (for iPhone and Android smart phones), Step Away chatbot, and assessment-only delay (control). Participants were assessed on their alcohol consumption and related behaviors when they enrolled in the study (baseline) and again 12 weeks later (fol low-up). Within the main study, we found that at 12 weeks participants in both intervention groups, reduced their drinking substantially, with large effect sizes noted. Participants in the waitlist control condition also reduced drinking substantially over the course of the study; results regarding the effectiveness of the app and chatbot relative to the control condition were thus inconclusive 20 “.   However, in the abstract of the article reference number 20, we can read “A total of 150 participants who completed the baseline and follow-up assessments were included in the final analysis.” Please describe the number of participants included in the final analysis.

-Lines 164-165: “Twenty semi-structured telephone interviews were conducted with participants who had completed follow-up assessments”. How did the authors realize that twenty interviews were sufficient for the study?

-Lines 174-177: “Of these, the 20 participants who used the bot or app the least, were assigned a number between 1 and 20, and the 20 participants who used the bot or app the most were also assigned a number between 1 and 20.” Please, this text is confusing.

RESULTS

·       Sample characteristics

-Please, the authors could write one paragraph about participant characteristics.

-Can the authors describe the educational level of the participants? The educational level may or may not facilitate the use of digital solutions.

·       Themes

How did the authors analyze the interviews? Did the authors perform a content analysis? Please describe.

LIMITATIONS

Lines 525-527: “There were a few participants who were randomly selected that did not respond to the interview recruitment.” How many?   This information is not available in the "Results" section.

REFERENCES

The authors could include some references from the year 2023.
